# Molecular Targeted Therapies: Time for a Paradigm Shift in Medulloblastoma Treatment?

**DOI:** 10.3390/cancers14020333

**Published:** 2022-01-11

**Authors:** Lidia Gatto, Enrico Franceschi, Alicia Tosoni, Vincenzo Di Nunno, Stefania Bartolini, Alba Ariela Brandes

**Affiliations:** 1Medical Oncology Department, Azienda Unità Sanitaria Locale, 40139 Bologna, Italy; lidia.gatto@ausl.bo.it (L.G.); vincenzo.dinunno@ausl.bologna.it (V.D.N.); 2IRCCS Istituto delle Scienze Neurologiche di Bologna, UOC Oncologia Medica del Sistema Nervoso, 40139 Bologna, Italy; a.tosoni@ausl.bologna.it (A.T.); stefania.bartolini@ausl.bologna.it (S.B.); alba.brandes@yahoo.it (A.A.B.)

**Keywords:** medulloblastoma, targeted therapy, Sonic Hedgehog (SHH), vismodegib, sonidegib, SHH pathway, SHH inhibitors, bromodomain proteins

## Abstract

**Simple Summary:**

In the last decade, medulloblastoma entered the molecular era, with progressive advances in the knowledge of the molecular biology of this rare tumor. These expanding data have allowed the recognition of four distinct molecular subgroups, and, subsequently, the design of novel clinical trials to update the treatment protocols adopted so far, with the introduction of new molecular targeted drugs.

**Abstract:**

Medulloblastoma is a rare malignancy of the posterior cranial fossa. Although until now considered a single disease, according to the current WHO classification, it is a heterogeneous tumor that comprises multiple molecularly defined subgroups, with distinct gene expression profiles, pathogenetic driver alterations, clinical behaviors and age at onset. Adult medulloblastoma, in particular, is considered a rarer “orphan” entity in neuro-oncology practice because while treatments have progressively evolved for the pediatric population, no practice-changing prospective, randomized clinical trials have been performed in adults. In this scenario, the toughest challenge is to transfer the advances in cancer genomics into new molecularly targeted therapeutics, to improve the prognosis of this neoplasm and the treatment-related toxicities. Herein, we focus on the recent advances in targeted therapy of medulloblastoma based on the new and deeper knowledge of disease biology.

## 1. Introduction

The standard treatment of medulloblastoma (MB) has not undergone substantial changes in the past 20 years and consists of a multimodal approach that includes surgery, craniospinal radiation and multi-agent chemotherapy [1].

However, these therapies are associated with both hematologic and non-hematologic side effects as well as severe and debilitating long-term complications, such as endocrinopathies, secondary malignancies, infertility and neurological impairments (sensory, motor and neurocognitive) [2,3].

The standard treatment allows for an average survival rate of 70%, whereas the remaining 30% of patients progress or relapse. In addition, adult MB is rare and can be considered an orphan pathology since therapeutic protocols are mainly extrapolated from experiences in children and only a few prospective and randomized studies have been, specifically, designed for the adult population [4,5,6,7,8,9].

For this reason, alternative personalized molecular therapeutic strategies specifically targeted to the biology of the tumor are needed in order to increase survival and to reduce both early and long-term toxicities [10].

From a molecular and clinical perspective, MB is a heterogeneous disease. Based on histopathological analysis, genome sequencing, methylation and phenotypic profiles, the International Consensus has proposed four distinct MB molecular subgroups: WNT activated, Sonic Hedgehog (SHH) activated, group 3 and group 4 [11,12].

MB-WNT is the most favorable group, whereas MB group 3 has the worst prognosis and MB group 4 has an intermediate prognosis [13]. SHH-activated MB, which accounts for approximately 30% of all MB cases, is prognostically “halfway” between the aforementioned subgroups, with a 5-year overall survival (OS) rate of about 50–70% in the absence of TP53 mutation [14]. SHH-activated and TP53 mutant MBs, instead, have a worse outcome than TP53-wild-type, presenting reduced PFS and a poor response to standard treatments and targeted therapies [15].

This molecular classification allows more accurate risk stratification of MB patients at diagnosis and has highlighted the prominent role of distinct deregulated signaling pathways, exploitable for a risk-adapted and personalized therapeutic approach. Nevertheless, no practice-changing prospective clinical trials have been published in adults to date.

## 2. Treatment of Shh Subgroup

The hedgehog (HH) signaling pathway is one of the key regulators of vertebrate embryonic organogenesis and homeostasis and is required for proper fetal and infantile tissue development and for the differentiation of hematopoietic, mammaryan and neural stem cells, particularly promoting the cerebellar maturation [15]. This pathway becomes normally inactive in adulthood, and its aberrant reactivation is responsible for the pathogenesis of MB [16].

The first knowledge on the HH signaling pathway was attributable to its close relation with Gorlin syndrome, a rare and hereditary disease arising around the age of 30, characterized by the development of basal cell carcinoma (BCC) of the skin and other different medical conditions, including changes and defects in the bone, cysts on the jaw, ovary fibromas, a rare tumor called rhabdomyosarcoma and MB [17,18].

In the absence of HH proteins (Sonic, Indian, or Desert Hedgehog), their ligand PTCH 1 inhibits the Smoothened gene (SMO) and acts as a negative regulator of the HH signaling pathway. Conversely, when HH proteins are present, they bind to the transmembrane receptor PTCH1, hindering its inhibition of SMO.

The activated transmembrane protein SMO is transferred to the primary cilium, a microtubule-based organelle of the cell surface, involved in extra-cellular and intra-cellular signaling pathways necessary for vertebrate development. When SMO is accumulated in the cilia membrane, the canonical (SMO-SUFU-GLI) and non-canonical (SMO-Gi-RhoA) signaling cascades are activated, both converging on the GLI (i.e., glioma-associated oncogenes) transcription network (Figure 1) [19,20]. GLI transcription factors translocate to the nucleus and upregulate BCL2, GLI1, HHIP, PTCH1, PTCH2 and other target genes that promote tumor proliferation, invasion, tumor metastasis and cancer stem cell survival [17,18].

Aberrant signaling in SHH-subgroup MB is due to genetic alterations, including PTCH1 loss of function a, SMO gain of function, SUFU loss of function, GLI1 or GLI 2 amplifications or MYC amplification [11,16,21,22].

In contrast to pediatric MB, approximately 60 to 70% of adult MB harbor an SHH-activated, TP53 wild-type molecular profile [23]. Adult MB is considered an orphan disease due to its rarity, with distinct clinical and molecular characteristics. In fact, in contrast to children, >80% of adult patients within the SHH subgroup predominantly harbor mutations upstream to SMO, particularly SMO or PTCH1 mutations, that have a very low mutation rate downstream to SMO [16,23]. These findings propose adult SHH tumors as an excellent candidate targetable by SMO inhibitors.

### 2.1. SMO Inhibitors

Great efforts have been made by pharmaceutical companies to develop small-molecule inhibitors of SMO, which can be used as selective antagonists of the HH pathway capable of directly binding to SMO, suppressing the downstream cascade.

SMO inhibition was first characterized in clinical trials performed with the topical application of cyclopamine, a natural steroidal alkaloid derived from Veratrum californicum, which demonstrated antitumor activity in basal cell carcinoma (BCC) [24]. Cyclopamine owes its name to the discovery of teratogenic effects in sheep, including fatal birth defects, holoprosencephaly and cyclopia.

Actually, three HH pathway inhibitors targeting the SMO receptor (vismodegib, sonidegib and glasdegib) have achieved approval for treatment of BCC or acute leukemia, but unfortunately, until now, these inhibitors have not been approved for the treatment of MB.

After FDA approval of vismodegib and sonidegib for metastatic BCC, SHH MB seemed to be the second ideal candidate for the development of SMO inhibitors [25]. Nevertheless, although initial preclinical and clinical studies documented encouraging efficacy of SMO inhibitors in the treatment of MB [26,27], sensitivity to SMO inhibitors likely resulted lower than that observed in BCC because of early drug resistance and low recruitment in clinical trials. Low accrual was observed, particularly in children, where SMO inhibitors induce a limiting side effect known as premature growth plate fusion. This toxicity, fortunately, is not relevant in adult patients, where growth plates have already fused [28].

Resistance to SMO inhibitors is caused by two mechanisms [29,30]:-Genetic alterations downstream to SMO that activate the SHH pathway (i.e., amplifications or mutations of SUFU and GLI) and the cross-talk between several molecular pathways, including Phosphoinositide-3 kinase (PI3K) and MEK. It has been proposed, for the future treatments of SHH-activated tumors, the use of combination therapies with SMO inhibitors and PI3K/MEK inhibitors [30], suggesting that concomitant treatment might increase the efficacy of SMO inhibitors-Mutations in SMO cause conformational changes in the SMO protein, hindering its binding to SMO inhibitors. Notably, a new generation of SMO antagonists, MK-4101, has been structurally designed to maintain efficacy even in vismodegib-resistant SHH MB [31].

Vismodegib is a synthetic SMO antagonist (Figure 1) belonging to the class of aryl amide that interacts with SMO via its 4-chloro-3-(pyridin-2-yl) aniline component, which was discovered using a high-throughput screen in vitro [32].

Sonidegib is an oral SMO inhibitor belonging to the class of biphenyl carboxamides that acts as a selective antagonist of the SMO (able to bind its drug-binding pocket), widely distributed within the tissues, with a very long half-life, effectively crossing the blood–brain barrier (BBB) [33,34]. After preclinical studies confirming the antitumor activity of vismodegib and sonidegib in MB [32,33], phase I [26,35,36,37] and phase II clinical trials of these SMO inhibitors were conducted [38,39] (Table 1).

The phase II trials, PBTC-032 and PBTC-025B, assessed the efficacy of vismodegib in pediatric (12 patients) and adult (31 patients) recurrent MB. Respectively, 1/12 children and 3/31 adults obtained objective responses. The benefit of prolonged PFS in adults was significantly longer in the SHH subgroup compared with the non-SHH subgroup. The authors observed that adult patients harboring mutations in either PTCH1 or SMO reported the most favorable outcomes, whereas patients with aberrations downstream to SMO, such as GLI or SUFU, did not respond, suggesting that SMO inhibitors efficacy depends on the position of mutations in the SHH pathway [36]. The results from these studies highlight that only a group of patients would benefit from SMO inhibitors according to patient age and molecular subtype.

Kian et al. have reported the case of an adult 24-year-old woman affected by desmoplastic MB, SHH-mutated and TP53 wild-type, harboring a non-sense mutation in position R135 of the PTCH1 gene, which achieved a remarkable and prolonged radiological response to the SMO antagonist vismodegib [41]. At the time of diagnosis, the MRI showed a wide right cerebellar and vermis lesion with multiple intramedullary metastases causing spinal cord compression. The patient underwent surgery for maximal safe resection and urgent decompressive laminectomy, followed by craniospinal radiotherapy. Although vismodegib is not approved for MB treatment, because of the patient aversion to chemotherapy and given the mutational profile, vismodegib was administered as the first-line therapy after craniospinal irradiation within a compassionate program after local institutional approval. Six months after starting vismodegib, the MRI revealed a complete radiological response, maintained until the moment of publication of the paper, approximately 15 months after starting vismodegib [37].

Recently, Frappaz et al. have reported the final results of the phase I/II MEVITEM study evaluating vismodegib + temozolomide vs. temozolomide alone in patients affected by recurrent/refractory SHH-activated MB [40] (Table 1). At the end of stage I, the study was prematurely terminated. Although the trial failed to demonstrate a 6-month progression-free survival (PFS) benefit in SHH recurrent/refractory MB, the combination of vismodegib and temozolomide was feasible without additional toxicity and obtained a significantly improved radiological response rate. The author concluded that there is a need to stratify patients more precisely according to molecular markers and that further studies are required to improve the handling of this targeted therapy.

Kresbach et al. have recently presented interesting data about the intraventricular administration of vismodegib. In a mouse model for SHH medulloblastoma, they have compared intraventricular and oral treatment with vismodegib, evaluating the effects on survival. Kaplan–Meier survival analysis showed a significant survival benefit with 1.6 mg/kg/d intraventricular vismodegib [42].

Sonidegib is a potent SMO antagonist, which showed efficacy in patients with solid tumors [35] and was evaluated in a phase II study [37] and in the phase II/III NCT01708174 trial in children and adults affected by MB.

Kieran et al. performed a phase I/II trial evaluating the safety profile and the tumor response in children and adults affected by relapsed MB [37]. The recommended sonidegib dose to be used for the phase II part of the study was fixed at 680 mg/m^2^ once daily for children and 800 mg for adults. On a total of 16 adult MB patients and 39 pediatric MB patients, the authors reported one partial response in an adult patient and four complete responses (two in the pediatric cohort and two in adults). The objective tumor responses were observed exclusively in SHH-tumors, particularly among the patients exhibiting a distinct five-gene SHH signature (downregulation of orthodenticle homeobox 2 and upregulation of GLI1, SHROOM2, sphingosine kinase-1 and PDLIM3). Conversely, patients lacking the five-gene signature did not respond [37,43]. Regarding toxicity, the most frequently reported adverse event with sonidegib in children was growth plate fusions that continued after cessation of therapy. In adults, instead, sonidegib demonstrated a favorable toxicity profile with rare hematological toxicities.

Sonidegib has also been recently investigated in the NCT01708174 trial, evaluating its safety and efficacy in adult and pediatric populations with SHH-activated relapsed MB. The study was initially designed as a randomized phase III study consisting of a randomized controlled part and a non-randomized uncontrolled part, in which 69 patients were to be randomized in a 2:1 ratio to receive sonidegib or temozolomide (TMZ). However, after the enrollment of 11 patients, the study was amended to a phase II single-arm study with only sonidegib, with a target enrollment reduced to 20 patients. The study enrolled two children who received sonideginb 500 mg/m^2^ orally and 16 adults who received sonidegib 600 mg orally. The primary endpoint of the study was the Overall Response Rate (ORR), which was 18.8% in adult patients and 0% in children. The secondary endpoint was PFS, which was 3.3 months in adults and 1.6 months in children.

The EORTC has recently proposed a new, unique multinational clinical trial (EORTC 1634-BTG/NOA-23) that will investigate the possibility to receive a reduced dose of craniospinal radiotherapy in SHH-activated newly diagnosed adult MBs in order to limit long-term side effects. In addition, the trial will explore the efficacy of the combination of the SMO inhibitor, sonidegib, in combination with the standard radio-chemotherapy treatment in this molecular subgroup [7].

Given the relevant long-term side effects of radiotherapy in MB patients, the aim of the study is to improve the efficacy and safety of future treatment protocols for MB, reducing radiotherapy-related toxicity. Patients will also be monitored for long-term quality of life and side effects of therapy, including cognitive impairment, infertility and social and professional limitations [7]. Of note, the study will randomize only adult patients, the ideal candidate for the clinical development of sonidegib.

The attention given to using SMO inhibitors in the adult population is also evidenced by another study, the ALLIANCE trial, which is still in development and aims to evaluate the antitumor activity of sonidegib (versus placebo) as maintenance therapy after radiotherapy in adults and young adults affected by newly diagnosed SHH-MB classified as standard/high-risk [44].

Li et al. have conducted a systemic review and meta-analysis among Phase I and II clinical trials involving MB patients treated with SMO antagonists. The clinical efficacy of SMO inhibitors was measured by ORR. They found that the pooled ORR of both SMO inhibitors, sonidegib and vismodegib, was 37% for SHH-activated MB but zero for other MB subtypes. Particularly, the pooled ORR of sonidegib was 55% among SHH-MB, whereas vismodegib produced a 17% ORR in the same population. Sonidegib performed significantly better than vismodegib in the pediatric population, producing the ORR 1.87- fold higher than that of vismodegib but demonstrated similar efficacy to vismodegib in the adult population [34].

In conclusion, vismodegib and sonidegib exhibit activity against adult recurrent SHH-MB but not against MB, as they lack the activation of the SHH pathway.

The main obstacles to the function of SMO inhibitors seem to be the widespread growth plate fusions observed in the pediatric population and the early onset of drug-resistance when the activation of the SHH pathway occurs downstream of SMO.

In conclusion, the SMO inhibitors vismodegib and sonidegib seem to be effective only in the subset of SHH-activated MB that harbors mutations upstream of SMO; furthermore, MB patients treated with SMO inhibitors exhibit treatment resistance and disease relapse over time, suggesting that therapy with the single agents alone may be inadequate, while combination therapies, especially with agents that are active downstream of SMO, might be an effective strategy to delay drug resistance and disease progression.

### 2.2. GLI Inhibitors

Activated GLI transcription factors translocate to the nucleus and upregulate target genes that control several processes of organogenesis, cell proliferation and survival. Here, we discuss GLI antagonists (GANTs): GANT-58 inhibiting GLI1 and GANT-61, an inhibitor of GLI1 and GLI2, both discovered by Lauth et al. from a cell-based screen for inhibitors of GLI1-mediated transcription using HEK293 cells [45]. GANT-61 has demonstrated the inhibition of GLI1/2 transcription factors in many cancers cell lines, including MB, rhabdomyosarcoma, neuroblastoma and osteosarcoma [46,47,48], which was more specific than GANT-58 in binding GLI proteins and more effective in reducing GLI1 and GLI2 activity. Nevertheless, because of GANT 61 instability at physiological conditions, no clinical trials are currently ongoing using GANT-61 to treat any cancer type [49].

### 2.3. Arsenic Trioxide (ATO)

Arsenic trioxide (ATO) antagonizes the canonical SHH pathway by directly binding to GLI1 and GLI2 with an IC50 of approximately 0.7 μM and received approval by FDA for the treatment of acute promyelocytic leukemia (used in combination with transretinoic acid therapy) [50,51]. ATO has also been shown to reduce cell viability, decrease clonogenic capacity and increase apoptosis rate in several SHH-activated tumor cell lines [52,53,54] and in derived MB allograft mice [51]. It is currently under investigation in several clinical trials, both alone in monotherapy and in combination with chemo- and radiotherapy.

### 2.4. Indirect Inhibitors of the Transcriptional Factor GLI: Histone Deacetylases (HDACs)

Histone deacetylases (HDACs) are epigenetic enzymes involved in controlling GLI1 and GLI2 by promoting their transcriptional activity through deacetylation [55].

HDACs are upregulated in SHH-activated MB; therefore, HDAC inhibitors have emerged as therapeutic agents to prevent the HH transcriptional cascade.

The dual epigenetic HDAC1/HDAC2 oral inhibitor mocetinostat has been demonstrated to successfully suppress tumor growth in SHH-MB mouse models [56].

Similarly, the selective HDAC6 inhibitor, ACY-1215 (rocilinostat), has been recently described as effective at reducing tumor growth in allografts of primary MB99–1 MB cells as well as of an MB99–1 allograft mouse model in vivo [57].

Panobinostat is a synthetic non-selective pan-deacetylase inhibitor that is currently under investigation in a pilot phase I study (NCT04315064) enrolling patients affected by recurrent or progressed MB. The study is aimed at investigating the association of panobinostat with MTX110, a gold-solubilizing nanoparticle. This compound can be directly administered intracranially, overcoming the BBB, thus achieving high brain bioavailability and limiting systemic side effects [58].

### 2.5. CDK Inhibitors

CDK4/6 inhibition is a candidate as a potential valid therapeutic option for the treatment of SHH and MYC-amplified group 3 MB. Cook Sangar et al., by informatics analysis and in vivo patient-derived xenograft models, demonstrated that the CDK4/6 pathway can be considered as a novel interesting target for all non-WNT MBs and that palbociclib, a highly selective cycline-inhibitor, causes cell cycle arrest in the G1 phase in xenograft models of MB [59,60,61]. The authors evaluated the efficacy of palbociclib as a single agent in mice bearing patient-derived xenografts of subcutaneous tumors that represented two different subgroups of MB: MYC-amplified group 3 and SHH mouse models [54]. They found significant therapeutic benefits of palbociclib in SHH-activated MB and also in MYC-amplified group 3 MB, which has the worst prognosis. They observed tumor regression with a reduction in tumor volume of 63% for both group 3-MYC amplified and SHH [59]. These data have been used by the Pediatric Brain Tumor Consortium to launch the Phase I trial NCT02255461, aimed at evaluating the safety of palbociclib in pediatric patients with recurrent or refractory brain tumors.

### 2.6. Bromodomain Proteins

Another class of epigenetic enzymes strongly involved in the control of the HH pathway is represented by bromodomain proteins. Bromodomains, comprised in the Bromodomain and Extra-Terminal Domain (BET) family of proteins, modulate the histone acetylation during cellular proliferation and differentiation and have a crucial role in the epigenetic control of oncogenes transcription [62]. Thus, BET inhibitors have been demonstrated to reduce the oncogenes’ expression and the GLI transcriptional activity [63].

Inhibitors of the bromodomain protein BRD4 have been shown to directly interact with GLI1 and GLI2 promoters, reducing the expression of HH target genes.

A small molecule named JQ1, an inhibitor of the bromodomain protein BRD4 by downregulating the transcription of GLI1 target genes, has been demonstrated to suppress cell proliferation in SHH MB cells resistant to SMO antagonists and to improve the survival of Ptch^+/−^-derived allografts of MB cells [64]. Recently, Wang et al. have demonstrated that the association of JQ1 with apolipoprotein E nanoparticles significantly improves the effectiveness of JQ1 in orthotopic MB-bearing mice [65].

Another BRD4 inhibitor, I-BET151, that dissociates BRD4 from the GLI1 locus remarkably suppresses tumor growth in Ptch^+/−^MB mice at a dose of 30 mg/kg [66].

### 2.7. CK2 Inhibitors

A novel interesting target downstream of SMO is CK2 inhibition [67]. Silmitasertib (CX-4945) is a novel oral small molecule that acts as a selective inhibitor of casein kinase II (CK2) and CK2-dependent hypoxia-induced factor 1 alpha (HIF-1α). CK2 is a protein kinase overexpressed in several cancer cells that stabilize GLI2, enhancing its transcriptional activity [68]. The NCT03904862 trial, a multicenter Phase I/II study testing the safety and tolerability of silmitasertib sodium in patients with recurrent SHH-activated MB, is still recruiting.

### 2.8. Lithium

A study by Zhukova et al. highlighted the role of lithium in the abrogation of TP53- related radiation resistance in SHH-activated MB [69]. The authors assumed that TP53 mutation is restricted to the SHH and WNT subgroups and is related to radioresistance and poor survival. However, TP53 mutation status is associated with poor survival in the SHH subgroup, whereas, in the WNT subgroup, it does not negatively influence survival and response to radiotherapy.

They hypothesized that the WNT pathway may play a crucial role in limiting the TP-53 related radioresistance and that this mechanism may be at the basis of the longer survival of patients among this subgroup. Lithium is an oral drug that acts as an activator of the WNT pathway. Zhukova et al. demonstrated that lithium increases the sensitivity of TP53 MB cells to radiotherapy, protecting healthy neural stem cells from the damage of radiotherapy, thus representing a valid therapeutic opportunity for high-risk MB [69].

## 3. Targeted Therapies for Wnt Subgroup

WNT tumors represent about 10% of all MBs and have the most favorable prognosis across all the subgroups [11,70,71,72,73]. WNT MBs are very sensitive to radiotherapy and chemotherapy, and most WNT patients survive after standard therapy.

WNT MBs are characterized by chromosome 6 monosomy either by somatic activating mutations of the CTNNB1 gene and have been reported germline loss-of-function mutations in the APC gene, associated with Turcot syndrome [74,75,76].

By increasing the knowledge of the molecular background of WNT MB, targeted therapies might be helpful to provide the best outcome with the least toxicity; however, targeting the WNT pathway is critical for several reasons:-WNT MB presents a leaky vascular endothelium that disrupts the blood–brain barrier integrity, increasing the penetration of chemotherapy, an important aspect that contributes to the excellent prognosis of this subgroup [75,77]. Thus, WNT signaling inactivation would eliminate the advantageous chemosensitivity of WNT-activated MBs.-WNT signaling is required for embryonal development and cell proliferation; thus, targeting this pathway would interfere with physiological mechanisms, such as tissue regeneration and normal development.-The WNT pathway has a crucial role in bone formation; thus, its inhibition can lead to osteoporosis [78].

Consequently, despite the deep understanding of the WNT pathway, therapeutic agents targeting WNT are not in clinical use yet. Only limited experiences on drugs that target β-catenin have been reported in preclinical models.

Interestingly, in 2012, Cimmino et al. reported the anticancer activity of cantharidin and norcantharidin—protein phosphatase inhibitors that promote the loss of nuclear β-catenin, both in MB cell lines and xenograft mouse models [79]. Cantharidin and norcantharidin demonstrated the ability to cross the blood–brain barrier, reduce β-catenin expression and inhibit the growth of intra-cerebellum tumors in orthotopic xenograft nude mice. N-(4-Hydroxyphenyl) retinamide (4-HPR, fenretinide), a synthetic analogue of all-trans-retinoic acids, has emerged as a valid compound targeting the WNT/β-catenin pathway in MB cells. It demonstrated the ability to inactivate β-catenin, inhibiting MB cell migration and invasion [80].

The new treatment protocols for this subgroup tumors are exploring whether low-risk WNT MBs can be treated with a lower dose of craniospinal radiation and a lower dose of chemotherapy to minimize side effects.

The EORTC 1634-BTG/NOA-23 trial will investigate if craniospinal de-escalation to 23.4 Gy (with boost) can be feasible and safe in WNT adult low-risk MB patients compared to the standard 35.2 Gy dose.

NCT01878617 is an ongoing phase II clinical trial evaluating the best therapeutic strategy for newly diagnosed MB by a molecular-based risk stratification. The aim of the study is to investigate whether patients with low-risk WNT tumors can be treated with a lower dose of craniospinal radiation and with a lower dose of cyclophosphamide to limit long-term side effects. Six weeks after radiotherapy, patients undergo four cycles of chemotherapy (cisplatin, vincristine, cyclophosphamide) with a lower dose of cyclophosphamide. NCT02724579 (COG ACNS1422) is another ongoing phase II trial exploring the possibility of adopting less-intense radiotherapy (craniospinal radiotherapy reduced at 18 Gray) and chemotherapy regimen (no vincristine during radiotherapy and reduced-dose maintenance chemotherapy) in WNT average-risk MB patients in order to avoid the late side effects of standard treatments.

The International Society of Pediatric Oncology (SIOP)’s study, “An International Prospective Study on Clinically Standard-risk Medulloblastoma in Children Older Than 3 to 5 Years With Low-risk Biological Profile (PNET 5 MB-LR) or Average-risk Biological Profile (PNET 5 MB-SR),” is the first trial stratifying MB patients into several risk-classes on the basis of clinical, histological and molecular factors. The name PNET 5 refers to the preceding PNET 4 study and is a trial designed to further improve the chances of treatment for patients with MB, minimizing the side effects that occur during and after standard treatment. In the low-risk arm, defined by localized WNT subgroup disease, the possibility of adopting a reduced craniospinal radiotherapy regimen (18 Gy) and a reduced maintenance chemotherapy regimen is under investigation. Two additional exploratory studies involving high-risk WNT and SHH-TP53 subgroups are designed to explore the optimal treatment for this rare under-studied subgroup [81].

## 4. Targeted Therapies for Group 3

Among the four subgroups, group 3 MB is the most aggressive with a dismal prognosis; therefore, it needs the development of novel targeted therapies.

Group 3 MB is characterized by the amplification/upregulation of the MYC oncogene, which entails an extremely poor prognosis, with a high rate of relapse and metastasis and scarce response to chemotherapy and radiation [82,83].

A key regulator of MYC expression is represented by epigenetic factors, such as HDACs, bromodomain inhibitors and SETD8.

A high-throughput screen performed by Pei and colleagues [84] identified HDAC inhibitors as potent therapeutic agents for group 3 MB. HDAC inhibition enhances the expression of FOXO1, an onco-suppressor gene, thus suppressing tumor growth [84].

PI3K inhibitors are also effective in inducing the expression of FOXO1 and act synergically with HDAC modulators, inhibiting tumor proliferation and prolonging survival in group 3 MB–bearing mice [84].

Bromodomain inhibitors, such as JQ1and BRD4, suppress the recruitment of transcriptional regulators of MYC, representing another promising avenue for effective therapy against group 3 MB [85]. SETD8 inhibitors target the chromatin occupancy at key genes involved in tumor growth of group 3-MYC mutated-MB [83].

Wang et al. reported SSRP1 as another promising target for epigenetic therapy against group 3-MYC mutated-MB [83].

Beyond MYC, CDK4/6 might also be a valuable target for group 3 MB, and the CDK4/6 inhibitor palbociclib has shown interesting activity in this subgroup [59].

## 5. Group 4 MB

Group 4 MB is the most common molecular subgroup, accounting for 35–40% of all MB cases [70]. It commonly involves children and adolescents, with a typical midline cerebellar location, is three times more frequent in males, and its prognosis, especially for average-risk patients, is good, with recurrences that tend to occur late, more than five years after diagnosis [70,86].

Interestingly, it has been identified a low-risk group of patients among metastatic group 4: chromosome 11 loss and/or chromosome 17 gain are prognostic chromosomal alterations that appear to dictate a more favorable prognosis, with 5-year PFS > 90%. No cytogenetic marker, instead, seems to be associated with poor prognosis [14,87,88,89].

The common somatic mutations are rare in group 4; thus, the development of targeted therapies has been rather scarce in this subgroup.

As in group 3 MB, amplifications in MYC and CDK6 can be considered driver events in group 4 tumors [90]; furthermore, recently, the homeobox transcription factor Lmx1A has been identified as an important transcriptional regulator in this subgroup [91].

Unfortunately, group 3 and group 4 have a low rate of somatic mutations [92,93], thus representing more disadvantaged subgroups in the search for an effective targeted therapy. In this context, some forward-looking technologies are emerging, capable of looking beyond DNA and gene expression, such as VIPER (virtual inference of protein activity by enriched regulon analysis), a new algorithm for accurate assessment of protein activity, useful for identifying the multiple dysregulated oncoproteins that contribute to tumorigenesis in a given patient in order to develop personalized treatments [94]. Similarly, DNA methylation analysis can also be a valuable strategy to identify subgroups of tumors potentially responsive to drugs that inhibit DNA methylation, such as brain tumors manifesting the cytosine-phosphate-guanine (CpG) island methylator phenotype (G-CIMP) [12,95].

## 6. Conclusions

Several emerging targeted compounds appear promising but further research efforts and the implementation of prospective clinical trials, including biomarker-led studies, are needed to recognize the molecular subgroups that can benefit from these innovative drugs, the most valid combination strategies and the appropriate treatment settings.

The most relevant molecularly stratified clinical trials have been designed for biological MB subgroups with more favorable prognoses; on the other hand, few efforts have been made to find effective targeted therapies for high-risk molecular subgroups, such as TP53-mutant SHH MB or metastatic/MYC-amplified group 3. Clinical–translational research should be encouraged to address this gap in order to provide the best survival and quality of life with the least toxicity for poor-prognosis MB patients. For example, it is a priority to study the mechanisms of leptomeningeal dissemination and to investigate the genes involved in this process and the suitable therapies to counter this metastatic capacity [49,96].

It is also imperative to support research for the identification of drugs that abrogate the detrimental effects of genetic events in the cancer therapeutic response and which can activate signaling pathways (such as the WNT pathway) capable of sensitizing high-risk MB to radiotherapy and chemotherapy [69].

Further work, mostly in the field of omics analysis, may be helpful to unveil novel molecular drivers and to improve the tailored treatment of MB and the quality of life of patients.

## Figures and Tables

**Figure 1 cancers-14-00333-f001:**
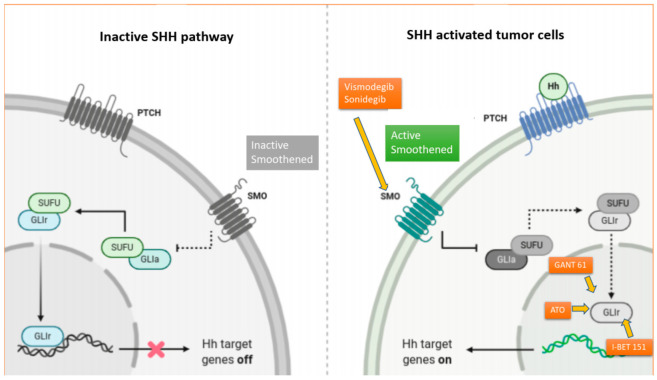
The complex Hedgehog signaling cascade and mechanism of action of SMO and GLI inhibitors. In the absence of HH proteins, PTCH 1 inhibits Smoothed (SMO) and acts as a negative regulator of the HH signaling pathway. Conversely, when HH proteins are present, they bind to PTCH1, hindering its inhibition of SMO. Activated SMO upregulates the SMO-SUFU-GLI signaling cascade, which promotes the transcription of the GLI (glioma-associated oncogenes), leading to tumor proliferation, invasion, tumor metastasis and cancer stem cell survival. Vismodegib and sonidegib are two small oral molecules that are selective antagonists of the HH pathway, which act by directly binding to SMO, inhibiting downstream activation of the HH-signaling cascade. GANT-61 is a GLI-inhibitor that has shown potent inhibition of GLI1 and GLI2 in preclinical studies of MB cancer cell lines. Arsenic trioxide (ATO) antagonizes the canonical SHH pathway by directly binding to GLI1 and has shown anticancer activity against MB both in in vitro and in vivo preclinical studies. I-BET 151, an inhibitor of the bromodomain protein BRD4, has also been shown to directly interact with GLI1 and GLI2, reducing the GLI-transcriptional activity.

**Table 1 cancers-14-00333-t001:** Completed clinical trials exploring targeted therapies for SHH-MB patients.

Trial	Author	Drug	Study Phase	Number of Mb Patients	Endpoints	Results
Phase I trial of hedgehog pathway inhibitor vismodegib (GDC-0449) in patients with refractory, locally advanced or metastatic solid tumors.	LoRusso 2011 [36]	Vismodegib	I	1	Safety and tumor responses	Acceptable safety profile. Antitumor activity was seen in 20/68 patients (19 with BCC and 1 MB).
Phase I study of vismodegib in children with recurrent or refractory medulloblastoma: a pediatric brain tumor consortium study.	Gajjar 2013 [37]	Vismodegib	I	33	Safety and tumor responses	Acceptable safety profile. Antitumor activity was seen in 1 of 3 patients with SHH-subtype disease.
Phase I, multicenter, open-label, first-in-human, dose-escalation study of the oral smoothened inhibitor sonidegib (LDE225) in patients with advanced solid tumors.	Rodon 2014 [35]	Sonidegib	I	9	Safety and tumor responses	Acceptable safety profile. Antitumor activity was seen in 6/16 patients with BCC and 3/9 patients with medulloblastoma (partial or complete response).
Phase II Clinical trial evaluating the efficacy and safety of GDC- II 0449 in adults with recurrent or refractory medulloblastoma.	Robinson 2015 [38]	Vismodegib	II	40	Safety and tumor responses	Acceptable safety profile. Antitumor activity was seen in 4/40 patients, all with SHH-subgroup MB.
Phase I study of oral sonidegib (LDE225) in pediatric brain and solid tumors and a phase II study in children and adults with relapsed medulloblastoma.	Kieran 2017 [39]	Sonidegib	I/II	55	Safety and tumor responses	Growth plate changes were observed in prepubertal pediatric patients. Antitumor activity was seen in 5 patients with SHH-subtype disease, 4 complete responders and 1 partial responder.
MEVITEM-a phase I/II trial of vismodegib + temozolomide vs. temozolomide in patients with recurrent/refractory medulloblastoma with Sonic Hedgehog pathway activation.	Frappaz 2021 [40]	Vismodegib + temozolomide	I/II	24	Safety and tumor responses	Terminated due to lack of success at the first stage of phase II.

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
