# Peer review of "Molecular Targeted Therapies: Time for a Paradigm Shift in Medulloblastoma Treatment?"

_cancers, 2022, doi:10.3390/cancers14020333_

Round 1

Reviewer 1 Report

  • the term post-pubertal is somewhat imprecise, as there is no exact cut-off and there are differences between males/females, further pediatric clinical trials often cover part of the post-pubertal phase as well; it would be more clear to use the term 'adulthood' were possible
  • there are some flaws in English language and style, the authors should revise the manuscript concerning this aspect, I give some hints in the following points
  • Group 4 is considered to have a intermediate prognosis and not the worst prognosis
  • Figure 1 is not clear: the inhibition of SMO is described to occur in the absence of HH protein, but yet is displayed in the 'on' phase; maybe it would be wise to include the inhibitors in the graphic as well
  • page 2: the prospective clinical trials published so far resulted in no...; this sentence is a little bit complicated
  • page 2: normally inactive in adulthood...
  • page 2: a rare and hereditary disease characterized by skeletal...
  • page 3: these inhibitors have not been approved...
  • page 3: the responses to these agents were variable and ...
  • page 4: please include that the MEVITEM trial was prematurely terminated and that the conclusion is that there is a need to stratify patients more precisely according to molecular genetic markers
  • page 6: is currently being investigated...
  • page 6: inhibitor that is currently being investigated...
  • page 7:  as a single agent in a subcutaneous xenograft mouse model representing two different...
  • page 7: please decide: Palbociclib or palbociclib
  • page 7: The NCT.... SHH-actived MB is still recruiting
  • page 7: completely regress after standard therapy
  • of course a table with the different agents, developmental stage (clinical trials), ability to cross the blood brain barrier or different application routes would be of help

Author Response

1)The term post-pubertal is somewhat imprecise, as there is no exact cut-off and there are differences between males/females, further pediatric clinical trials often cover part of the post-pubertal phase as well; it would be more clear to use the term 'adulthood' were possible

  • Author response: We thank the reviewer for this suggestion. We have replaced the term "post-pubertal" with “adulthood” throughout the manuscript. You can find it in yellow in the text.

2)There are some flaws in English language and style, the authors should revise the manuscript concerning this aspect, I give some hints in the following points

  • Author response: Thanks for this comment. We have modified the text by correcting the grammar and the style. You can find it in yellow in the manuscript.

3)Group 4 is considered to have an intermediate prognosis and not the worst prognosis

  • Author response: We appreciate this critical comment. As suggested, we have modified this piece of information in the text. You can find it in yellow on page 19.

4)Figure 1 is not clear: the inhibition of SMO is described to occur in the absence of HH protein, but yet is displayed in the 'on' phase; maybe it would be wise to include the inhibitors in the graphic as well

  • Author response: Thanks for this suggestion. We have modified the Figure 1.

5) Please include that the MEVITEM trial was prematurely terminated and that the conclusion is that there is a need to stratify patients more precisely according to molecular genetic markers

  • Author response: Thanks for this comment. As suggested, we have included this piece of information into the text. You can find it in yellow on page 8.

6)Of course, a table with the different agents, developmental stage (clinical trials), ability to cross the blood brain barrier or different application routes would be of help

Author response: We thank the reviewer for this comment. As suggested, we have added a table in the manuscript (Table 1), which describes the completed clinical trials exploring targeted therapies for SHH-MB patients

Reviewer 2 Report

The authors have reviewed the literature on medulloblastoma with a focus on potential  molecular targeted therapies according to subgroups and molecular characteristics. This is a comprehensive review, although some sections are relatively limited (for example “targeted therapies for WNT subgroup). One important question relates to the potential interest of next generation sequencing, considering the low likelihood of finding clinical actionable alterations.

Specific comments:

  • The authors should review the text carefully. Somme sentences need to be clarified or rewritten, such as: “Nevertheless, unfortunately, the prospective clinical, until now published, resulted no practice-changing and, to date, no targeted therapies are approved for the treatment of MB.” or “the responses to these agents re-sulted variable and, generally, much lower than those observed in BCC” or “although the maintenance of the response resulted modest, no longer than 4 months”
  • The section on SMO inhibitors is very detailed. The authors could add a comment on the recent presentation of Kresbach et al on the intraventricular administration of vismodegib (Neuro-oncology 2021, Vol23, supplement 6, page vi179 – abstract)
  • Another suggestion is the addition of a small section on TP53 mutant SHH medulloblastoma and the work on lithium that suggests a potential therapeutic benefit in combination with radiotherapy (Zhukova et al, PMID: 25539912)
  • The section on WNT is relatedly disappointing and does not provide any comment on the future of WNT inhibitors.
  • The section on Group 4 has some inaccuracies. The data from Shih et al (PMID: 24493713) and 2 recent clinical trials (ACNS0331 and SJMB03, PMID: 34110925 and 33405951, respectively) suggest a much higher survival rate for patients with chromosome 11 loss, in the range of 90% at 5 and 10 years, and no impact of the presence of isochromosome 17q on survival (see supplemental figure of the 2 trials)

Overall, this is a comprehensive review of current knowledge. However, this review also illustrates the disconnection between current knowledge and ongoing clinical trials. While we know that current protocols for very high risk medulloblastoma (TP53 mutant SHH or metastatic group 3 MYC amplified tumors for example), no attempt has been made – with the exception of the PNET5 trial that has an specific recommendation for TP53 SHH mutant tumors – to alter the treatment of such patient. A comment on potential strategies to address these gaps would be important.

Author Response

1)The authors should review the text carefully. Somme sentences need to be clarified or rewritten, such as: “Nevertheless, unfortunately, the prospective clinical, until now published, resulted no practice-changing and, to date, no targeted therapies are approved for the treatment of MB.” or “the responses to these agents resulted variable and, generally, much lower than those observed in BCC” or “although the maintenance of the response resulted modest, no longer than 4 months”

  • Author response: We thank the reviewer for the revision. We have modified the text by correcting the grammar and the style. You can find it in yellow in the text.

2) The section on SMO inhibitors is very detailed. The authors could add a comment on the recent presentation of Kresbach et al on the intraventricular administration of vismodegib (Neuro-oncology 2021, Vol23, supplement 6, page vi179 – abstract)

  • Author response: We appreciate this critical comment. We have included a short discussion concerning this presentation in the section named “SMO inhibitors”. You can find it in yellow on page 8.

3) Another suggestion is the addition of a small section on TP53 mutant SHH medulloblastoma and the work on lithium that suggests a potential therapeutic benefit in combination with radiotherapy (Zhukova et al, PMID: 25539912)

  • Author response: We thank the reviewer for raising this issue. We have added a paragraph named “Lithium”. You can find it in yellow on page 15.

4)The section on WNT is relatedly disappointing and does not provide any comment on the future of WNT inhibitors.

  • Author response: We thank the reviewer for the revision. We have expanded the section on WNT medulloblastoma and WNT inhibitors. We have included this information into the text in yellow, on pages 16-18.

5)The section on Group 4 has some inaccuracies. The data from Shih et al (PMID: 24493713) and 2 recent clinical trials (ACNS0331 and SJMB03, PMID: 34110925 and 33405951, respectively) suggest a much higher survival rate for patients with chromosome 11 loss, in the range of 90% at 5 and 10 years, and no impact of the presence of isochromosome 17q on survival (see supplemental figure of the 2 trials)

  • Author response: We thank the reviewer for the revision. We have included this piece of information into the text in the paragraph named “GROUP 4 MB”. You can find it in yellow on page 19.

6)Overall, this is a comprehensive review of current knowledge. However, this review also illustrates the disconnection between current knowledge and ongoing clinical trials. While we know that current protocols for very high risk medulloblastoma (TP53 mutant SHH or metastatic group 3 MYC amplified tumors for example), no attempt has been made – with the exception of the PNET5 trial that has a specific recommendation for TP53 SHH mutant tumors – to alter the treatment of such patient.  Comment on potential strategies to address these gaps would be important.

Author response: Thank you very much for raising this critical point. We have included a comment on potential strategies to address high risk medulloblastoma into the text, in the paragraph named “Conclusions”. You can find it in yellow on pages 20-21.

Reviewer 3 Report

Medulloblastoma is a rare malignancy and not much research has been done in case of adults.The authors have done a creditable work by compiling this review which will benefit clinicians and researchers working in this field.

The molecular subtypes of Medulloblastoma are categorized into 4 groups.All the 4 groups are described well independently and bring insight for the researchers.

The targeted therapies for these 4 groups have been detailed very well and provide useful information.

Author Response

Dear Reviewer thank you very much for your support and revision.